10

20

# Quantitative uncertainty and post-processing for microaethalometers measuring black carbon

Timothy A. Sipkens<sup>1</sup>, Joel C. Corbin<sup>1</sup>, Kerry Chen<sup>3</sup>, Laura-Helena Rivellini<sup>2</sup>, Jonathan Abbatt<sup>2</sup>, Jason S. Olfert<sup>3</sup>

- <sup>1</sup>Metrology Research Centre, National Research Council Canada, Ottawa ON Canada
  - <sup>2</sup>Department of Chemistry, University of Toronto, Toronto ON M5S 3H6, Canada
  - <sup>3</sup>Department of Mechanical Engineering, University of Alberta, Edmonton AB Canada

Correspondence to: Timothy A. Sipkens (Timothy.Sipkens@nrc-cnrc.gc.ca)

Abstract. Aethalometers measure black carbon mass concentrations by monitoring light attenuation through a particle filter as it becomes laden with aerosols. As the uncertainties in the resulting measurements are not easily quantified via a bottom-up traceable approach, there is a need for inter-device comparisons to provide operationally defined uncertainties. The present work compared five micro-aethalometers to known mass concentrations of laboratory-generated soot, formed using an inverted ethylene flame and a Centrifugal Particle Mass Analyzer-Electrometer Reference Mass Standard (CERMS). Uncertainties were found to scale with mass concentration, with multiplicative errors between devices of approximately 10 % in the best case of long sampling times and/or high mass concentrations. A quantitative expression is provided for the uncertainty in the aethalometer measurements as a function of mass concentration, sampling interval, and flow rate. An open-source algorithm is also provided for the unsupervised reanalysis of aethalometer or other filter photometer data over varying periods to reach a specified target uncertainty.

#### 1. Introduction

Typical filter photometers measure the light attenuation of a particle-laden filter, from which a reasonably accurate measurement of aerosol light absorption or equivalent mass can be derived (Lack et al., 2014; Moosmüller et al., 2009). Such light absorption measurements are obtained by assuming a proportionality between light attenuation and absorption, which can then be used to estimate the radiative properties of the atmosphere. Alternatively, by assuming a proportionality between light absorption and mass (i.e., a mass absorption cross-section,  $\sigma_{MAC}$ ) photometer data can be used to estimate equivalent black-carbon mass (eBC) (Petzold et al., 2013) for use in human exposure assessments (Weichenthal et al., 2015). These assumed constants of proportionality are in fact subject to variability with particle size and mixing state (Nakayama et al., 2010; Drinovec et al., 2015) as well as spurious eBC signals when attenuation is caused by light scattering rather than absorption (Drinovec et al., 2015). Nevertheless, filter photometers remain widely used due to their low cost, low weight (advantageous for both aircraft and personal exposure studies), and ability to measure at multiple wavelengths (Chakraborty et al., 2023; Pikridas et al., 2019; Caubel et al., 2018).

50

55

The accuracy of filter photometers has been evaluated by multiple studies. It is well known that the proportionality between filter attenuation and particulate light absorption is sensitive to the absolute attenuation of the filter, which has been formulated in terms of correction factors – such as an empirical correction factor, *R* (Weingartner et al., 2003), and a correction factor for loading, *k* (Virkkula et al., 2007) – using various algorithms, some of which take into account scattering cross-sensitivities (Collaud Coen et al., 2010). This issue has been addressed recently by dual-spot photometers which sample in duplicate at different flow rates (Drinovec et al., 2015). Accuracy may also be affected by particle size and morphology. Nakayama et al. (2010) measured the response of a particle soot absorption photometer (PSAP) and a continuous soot monitoring system (COSMOS) to nigrosine aerosols of different sizes. Those authors showed that the attenuation-to-absorption conversion function depends on particle size. The spherical morphology of nigrosine allowed those authors to use Mie theory as a reference. In contrast, for AE33 aethalometers, Drinovec et al. (2015) used nephelometer measurements as a reference and reported that attenuation-to-absorption conversion did not depend greatly on particle size. These correction factors are not perfect. For example, rapid changes in gas-phase composition such as humidity spikes can also lead to measurement biases, as has been shown for three different filter photometer models (AE51, MA200, and PSAP) (Düsing et al., 2019; Cai et al., 2014; Arnott et al., 2003).

Independently, the mass absorption cross-section  $\sigma_{MAC}$  is known to vary between samples. For example, the  $\sigma_{MAC}$  of soot at 550 nm wavelength for mass-integrated samples may vary from  $8.0 \pm 0.7$  m<sup>2</sup> g<sup>-1</sup> (Liu et al., 2020) up to about 14 m<sup>2</sup> g<sup>-1</sup> due to intraparticle scattering ("lensing effect") upon condensation of semi-volatile materials on soot (Cappa et al., 2019; Fierce et al., 2020). Extremely small soot particles may also have smaller MACs (Corbin et al., 2020). The conversion of light absorption to eBC is also complicated by the presence of light-absorbing carbon other than soot, including soluble brown carbon and tar brown carbon ("tarballs"), which can absorb even in the infrared (McMeeking et al., 2014; Chakrabarty et al., 2023; Cheng et al., 2024; Corbin et al., 2019). Nevertheless, these issues reflect the complexity of atmospheric aerosols rather than limitations of the filter-based approach.

In addition to the fundamental physical uncertainties discussed above, uncertainties for filter photometers may also be considered in terms of measurement reproducibility between instruments. Cuesta-Mosquera et al. (2021) compared 23 dual-spot Magee AE33 aethalometers and reported roughly  $\pm$  10% reproducibility between instruments. Chakraborty et al. (2023) compared three AethLabs MA300 micro-aethalometers with an AE33 aethalometer and reported strong correlations of  $R^2 = 0.9$ . The latter study did not report its results in terms of between-instrument reproducibility, and both studies used other aethalometers as references, rather than reference measurements of eBC concentration or light absorption.

The intercomparison studies summarized above report overall comparisons, but do not present quantitative uncertainty models representing between-instrument reproducibility for aethalometers. Such models allow for uncertainty evaluations under such scenarios as changing sample flow rate or mass concentrations. When developed using reference measurements of mass concentration (as done here) or light absorption, such models also allow for the evaluation of aethalometer biases, in contrast to aethalometer-only intercomparisons. In this study, we report a laboratory characterization of the reproducibility of five micro-aethalometers (Aethlabs MA200 and MA300) using non-volatile soot aerosols. We quantify reproducibility

between devices and between filter changes and formulate a simple error model for uncertainty estimation in future micro-aethalometer measurements. A corollary of this error model is that it can be used to estimate the optimal aethalometer sampling time for a given mass concentration. An algorithm to implement this optimal sampling time, based on earlier work by Hagler et al. (2011) is presented. Since our aerosol model represents a simple source of eBC, with negligible content of non-absorbing PM and stable gas-phase composition, our results provide a lower limit on between-instrument reproducibility.

## 2. Experimental setup

We tested five microAeths (two MA200, three MA300, AethLabs, USA) that were previously deployed at ambient monitoring sites across Canada. The aethalometers were collected and placed in a metal chamber with a circulation fan to ensure even mixing for testing. The equivalent black carbon (eBC) mass concentration from the aethalometers is derived from (Drinovec et al., 2015)

$$M = \frac{\text{d ATN}}{\text{d}t} \frac{1}{Q_a} \frac{S}{\sigma_{\text{MAC}} \cdot C} \left( \frac{1}{1 - k_c \cdot \text{ATN}} \right), \tag{1}$$

where M is the eBC mass concentration; ATN is the attenuation through the filter; t is time;  $Q_a$  is the flow rate through the aethalometer filter, after accounting for leakage; S is the cross-sectional area of the filter;  $\sigma_{MAC}$  is the mass absorption cross section or coefficient (m²/g); C is an optical absorption enhancement factor, accounting for apparent enhancement of the absorption, mostly due to light scattering by both the particles and filter; and  $k_c$  is the dual spot correction factor accounting for attenuation effects on the filter, which is defined in (Drinovec et al., 2015). The correction factor is taken as that provided by the instrument firmware and applied during post-processing. In practice, the mass concentration is computed over a sample interval and the attenuation mass attenuation cross section is rewritten in terms of particle and filter properties,

$$M \approx \frac{\Delta ATN/100}{\Delta t} \frac{1}{Q_a} \frac{S}{\sigma_{MAC} \cdot C} \left( \frac{1}{1 - k_c \cdot ATN} \right), \tag{2}$$

where  $\Delta t$  is the sampling interval and  $\Delta$ ATN is the change in the attenuation over that interval, with the factor of 100 allowing for consistency with the original definition of Gundel et al. (1984). Initially, we consider the minimum sampling interval of  $\Delta t$  = 10 s.

It is further noted that Eq. (2) can be phrased in terms of a change of particulate mass  $\Delta m$  on the filter in a given time interval, where

$$M = \frac{\Delta m}{Q_a \cdot \Delta t} , \qquad (3)$$

such that

100

105

$$\Delta m = \frac{\Delta ATN}{100} \cdot \frac{S}{\sigma} \frac{1}{1 - k_c \cdot ATN} \tag{4}$$

where  $\Delta m$  is the mass collected on the filter in each time interval. This formulation of the aethalometer response allows for the uncertainty associated with sample flow and sampling duration to be considered independently from uncertainties associated with the amount of mass loaded onto the filter. It also results in fundamentally meaningful units of mass for the analysis below.

For testing, the aethalometer chamber was periodically filled with soot, after which the inlet to the chamber was closed and the concentration of particles in the chamber was allowed to decay slowly over time, as shown in Figure 1. BC particles were generated using a Mini Inverted Soot Generator operated on ethylene (MISG; Argonaut Scientific; (Kazemimanesh et al., 2019)). The particle stream was passed through a unipolar charger (Unipolar Diffusion Aerosol Charger, UDAC; Cambustion Ltd, UK) with an ion-concentration time product of  $5x10^{12}$  s/m³, and a Centrifugal Particle Mass Analyzer (CPMA; Mk II, Cambustion; (Olfert and Collings, 2005)), which selects particles of a narrow range of mass-to-charge ratios. Particles are then directed to the experimental chamber. Flow leaving the experimental chamber is passed to a Faraday cup aerosol electrometer (FCAE; 3068B, TSI), which measures the total current. The flow rate in the FCAE was controlled with an external mass flow controller (MCP-10SLPM-D/5M, Alicat Scientific, USA). Given that the particles were previously classified by the CPMA and thus have a fixed mass-to-charge ratio, the mass concentration in the chamber can be computed as

$$M = \frac{m^* I}{eO_0} \,, \tag{5}$$

where M is the mass concentration (µg/m³),  $m^*$  is the CPMA mass-to-charge setpoint (fg), I is the measured electrometer current (fC/s), e = 0.1602 aC is the elementary charge, and  $Q_c$  is the flow rate through the FCAE (cm³/s) (Symonds et al, 2013). For all experiments, the CPMA was set to a single setpoint, a mass-to-charge setpoint of  $m^* = 0.4$  fg/e and a resolution of  $R_m = 5$ . The FCAE flow rate was fixed at  $Q_c = 1$  L/min (with temperature and pressure near ambient conditions).

To calculate the amount of mass expected to be collected on the aethalometer filter, we applied Eq. (3) to the reference mass concentration from the CERMS,

$$\Delta m_{\rm c} = Q_{\rm a} \cdot \Delta t \cdot \left( \frac{m^* I}{e Q_{\rm c}} \right) \,, \tag{6}$$

where  $\Delta t$  and  $Q_a$  correspond to the aforementioned aethalometer sampling interval and flow rate. Note, if  $Q_a \cdot \Delta t$  is given in mL and the mass concentration in  $\mu g/m^3$ ,  $\Delta m$  will be returned in pg.

A scanning mobility particle sizer (SMPS) – composed of an electrostatic classifier (3082, TSI), a differential mobility analyzer (3081, TSI), and an Ultrafine Condensation Particle Counter (3776, TSI) – was situated downstream and indicated that the size distribution had a geometric mean mobility diameter of 235 nm and geometric standard deviation of 1.53. This

distribution was unimodal, and did not show multiple-charging effects, because the multiple charges imparted to each particle from the UDAC results in blurring of such effects (Sipkens et al., 2023).

Figure 1. Experimental setup wherein a reference mass produced by the CERMS is used to feed a known mass concentration of soot particles into chamber containing a series of aethalometers.

#### 3. Results and discussion

Figure 2 shows data from one of the aethalometers (Device 1). Measurements clearly show the periodic nature of the measurements, where the aethalometer chamber is filled with particles, following which the concentration in the chamber is allowed to decay over time. Two instances of the tape changing in the aethalometer are also visible when the mass concentrations were high. Only dual-spot corrected data is used in the analysis (i.e., when ATN > 3, as measurements of 3 


Figure 2. Mass concentrations reported by a single aethalometer (Device 1) at a sampling interval of 10 s. The periodic filling of the box with particles is clearly visible, as well as the change of the tape for two cases where the mass concentrations in the box were high. (c) A subset of the data from Day 2, focusing on data around a tape change, including data where ATN < 3. Inspection shows the reduction in the noise when ATN < 3, where the dual-spot algorithm is not yet applied to the data.

## 3.1. Types of variability in aethalometer data

Figure 3 shows scatter plots of the mass collected by the aethalometer (over 10 s sampling intervals, corresponding roughly to 5-minute sampling intervals at  $M = 0.3 \,\mu g$  m<sup>-3</sup>; Eq. (4)) against that measured by the reference, as well as the difference and ratio between the two measurements. The collected mass is correlated as expected, roughly distributed about the line of parity. In this subsection, we first make several observations from these plots before proposing a quantitative aethalometer uncertainty model in the next subsection.



Figure 3. Comparison of eBC mass reported by the aethalometers to reference (CERMS) mass measurements. The upper row indicates parity plots, (a) using a linear scale and (b) using a log scale. Bottom panels show (c) the difference between the reference and aethalometer mass and (d) percentage difference between the CERMS reference and aethalometer mass concentrations, calculated as  $\Delta m_a / \Delta m_c - 1$ . Data have been thinned by a factor of three (every third data point is plotted) for visualization. Measurements taken when the attenuation was below 3 are excluded. Green annotations indicate model fits derived later in this work.

(1) At low mass concentrations, variability in the measurements exhibited a consistent spread on a linear scale (cf. the left portion of Figure 3c where  $\Delta m_c 






contribute considerable uncertainties, the correction is essential to ensuring that a correct (unbiased) mass concentration is returned, as has been well established (Drinovec et al., 2015).

(2) As the mass concentration increases, the absolute error expands (cf. Figure 3c), while the relative error decreases (cf. Figure 3d). The growth in the errors seems consistent with Poisson noise and such noise is a logical addition. Poisson noise would result when considering the discrete arrival of particles at the filter over a given interval. As the particle counts should be high, the noise should be well-approximated as Gaussian but with a variance that scales with collected mass. Mathematically,

$$s_{\rm p}^2 = p \cdot \Delta m \ , \tag{7}$$

where  $s_p^2$  is the variance due to Poisson noise and p is a factor indicating the significance of the Poisson noise, and stems from the fact that raw counts would be scaled to obtain a mass.

(3) While not explicit in Figure 3, a second source of Poisson noise could stem from photon shot noise in the detector. For a given sampling interval, this noise will not vary significantly across the observations. However, unlike the previous source of Poisson noise, this source of Poisson noise will increase with the sampling interval, as more photons will be collected by the detector. Combined with (1) and (2), this would suggest that the random scatter in Figure 3 (and, by extension, Figure 5 later in this work) can be modeled as

$$var(e) = (p \cdot \Delta m + \gamma^2)(\Delta t/\Delta t_0) , \qquad (8)$$

where e denotes the random measurement error, which constitutes an error term under repeatability conditions (i.e., the noise observed when sampling a stable aerosol);  $\Delta t$  is the sampling interval; and  $\Delta t_0$  is a reference sampling interval that acts to normalize the contribution ( $\Delta t_0 = 10$  s in the current work).

- (4) Device effects (i.e., biases for a specific device) present as multiplicative errors. Such errors manifest in a difference between the aethalometers and reference mass that scale with mass and a ratio that has a roughly constant spread (cf. Figure 3b and d). This choice also matches the observations of previous studies of Magee AE31 Aethalometers (Cuesta-Mosquera et al., 2020). Due to the multiplicative nature of the device effects, they are not discernible at low mass concentrations, where the Gaussian and Poisson noise terms dominate. However, at high mass concentrations, there is some stratification where some aethalometers measured higher than others and vice versa. The multiplicative nature of the inter-device error is an indication of an error that comes from fluctuations in a contribution that is multiplied by the quantity-of-interest (e.g., flow rate discrepancies, or filter leakage terms; see also (Drinovec et al., 2015)).
- (5) There is a systematic, additive bias in the measurements at low mass concentrations, manifesting as an offset in the y-axis that is particularly visible in Figure 3c (horizontal green line). Reanalysis of the data showed that this bias is directly proportional to the sampling interval  $\Delta t$  in Eq. (6). This offset has not been identified in previous aethalometer intercomparisons that included only filter photometers (Cuesta-Mosquera et al., 2020; Chakraborty et al., 2023), or in



which the analysis forced fits through the intercept (Cuesta-Mosquera et al., 2021). In our data, this bias seems to be consistent across all the measurements, that is the aethalometers themselves are self-consistent, which remains consistent with previous work. A common bias across aethalometers could be caused by biases in reference mass concentrations. The precise cause of this offset is not currently understood and will be treated naively as a bias in the model.

- (6) Inaccuracy in any of the multiplicative terms in Eq. (2), including the MAC, flow rate (or leakage of the aerosol flow), and loading correction would manifest as a systematic, multiplicative error. In other words, if the MAC used by the device is 10% larger than it should be, this would contribute to the aethalometer-derived mass concentrations being 10% lower than CERMS mass concentrations across the domain. This kind of error is different to those described in (1) and (2) above, as a fixed inaccuracy results in a systematic bias, not random noise. Fitting a model with this kind of multiplicative bias indicated this effect was negligible for the IR channel, such that this value was not considered in subsequent analysis. This would suggest that the applied conversion to mass concentration by the device is reasonable.
- (7) While not shown in Figure 3, it is noted that the attenuation coefficient has minimal effect above an attenuation of 3, that is following an initial period after the tape changed. Below this point, the data was not corrected (cf. Figure 2) such that fitting was performed neglecting these data.

These observations support a model for the aethalometer response of the form:

$$\Delta m_{a}(j) = \Delta m_{c} + \beta \cdot [Q_{a} \cdot \Delta t] + l_{j} \cdot \Delta m_{c} + e(\Delta m_{c})$$

$$\tag{9}$$

where  $\Delta m_a(j)$  is an aethalometer measurement from the *j*th device;  $\Delta m_c$  is the expected change in mass on the filter, computed here from the reference (CERMS) mass concentrations;  $\beta$  is an additive bias in the measurements, accounting for the *y*-offset observed in the parity plots, and is a function of  $Q_a$  and  $\Delta t$  according to Eq. (8);  $l_j$  is a device-specific bias, analogous to the laboratory effect in an interlaboratory study (ISO, 2019); and  $e(\Delta m_c)$  is the measurement error defined in Eq. (8).

## 210 3.2. Model fitting and uncertainty quantification

The model is fit within the Bayesian framework, which encodes prior knowledge in terms of probability distributions. The framework is based on Bayes' equation

$$p_{po}(\mathbf{x}, \boldsymbol{\theta} \mid \mathbf{b}) \propto p_{li}(\mathbf{b} \mid \mathbf{x}, \boldsymbol{\theta}) p_{pr}(\mathbf{x}, \boldsymbol{\theta})$$
, (10)

or, in logarithmic form,

$$\ln p_{po}(\mathbf{x}, \boldsymbol{\theta} \mid \mathbf{b}) = \ln p_{li}(\mathbf{b} \mid \mathbf{x}, \boldsymbol{\theta}) + \ln p_{pr}(\mathbf{x}, \boldsymbol{\theta}) + C_0 , \qquad (11)$$

where  $p_{li}(\mathbf{b}|\mathbf{x},\mathbf{\theta})$  is the likelihood, used to relate the observations to the mixed effects model;  $p_{pr}(\mathbf{x},\mathbf{\theta})$  is the prior, describing 215 information known about the parameters a priori;  $p_{po}(\mathbf{x}, \boldsymbol{\theta}|\mathbf{b})$  is the posterior, which is used to compute the expected value of and uncertainties in the various model parameters; and  $C_0$  is a constant to yield a properly scaled posterior probability density function. Within this formulation, there are added nuisance parameters,  $\theta = [s_1^2, \gamma, p]$ , which here corresponds to the unknown inter-device variance,  $s_1^2$ , and the error model parameters from Eq. (8). These quantities are inferred alongside the effects in x.

In order to fit the uncertainty model given in Eq. (9), we arrange each effect into a vector,

$$\mathbf{x} = [\beta, l_1, l_2, l_3, l_4, l_5]^{\mathrm{T}} , \tag{12}$$

where the five device-specific biases  $l_i$  correspond to the five devices considered in this study, and  $\beta$  corresponds to any bias consistently observed between the aethalometers and the reference. We then arrange the measurement data as a vector as b,

$$\mathbf{b} = \begin{bmatrix} b_1, b_2, \dots \end{bmatrix}^{\mathrm{T}} = \begin{bmatrix} \Delta m_{\mathbf{a},1}(j_1) - \Delta m_{\mathbf{c},1}, \Delta m_{\mathbf{a},1}(j_2) - \Delta m_{\mathbf{c},2}, \dots \end{bmatrix}^{\mathrm{T}}, \tag{13}$$

where  $\Delta m_{a,i}(j_i)$  is the *i*th  $\Delta m_a$  measurement made with the  $j_i$ th device. The matrices **x** and **b** may be related one another by a design matrix, **D**, and a random-error matrix, **e**, such that

$$\mathbf{b} = \mathbf{D}\mathbf{x} + \mathbf{e} \tag{14}$$

where e compiles the random measurement errors. Eq. (14) is formulated such that the *i*th row in **D**, denoted as  $\mathbf{d}^i$ , corresponds 225 to a single measurement, and will be mostly zero, since the effects in  $\mathbf{x}$  are device specific. For example, the  $\mathbf{d}^i$  corresponding to any measurement corresponding to the 1st device would be,

$$\mathbf{d}^{i} = [Q_{\mathbf{a}} \cdot \Delta t, \Delta m_{\mathbf{c}}, 0, 0, 0, 0], \tag{15}$$

where the detailed uncertainty terms representing Poisson, Gaussian, and multiplicative noise discussed in Section 3.1 are represented within  $\Delta m_c$ .

We fit the error terms in Eq. (15) as follows. Errors in the measurements are assumed to be Gaussian distributed with a 230 Poisson-Gaussian variance following Eq. (8). It can then be shown that the likelihood relating the data to the error model is,

$$\ln p_{li}\left(\mathbf{b} \mid \mathbf{x}, \mathbf{\theta}\right) = -\frac{1}{2} \sum_{i} \ln \left[ \left( pM_{i} + \gamma^{2} \right) \left( \Delta t / \Delta t_{0} \right) \right] - \frac{1}{2} \sum_{i} \frac{\left(\mathbf{d}^{i} \mathbf{x} - b_{i}\right)^{2}}{\left( pM_{i} + \gamma^{2} \right) \left( \Delta t / \Delta t_{0} \right)} + C_{0} . \tag{16}$$

The latter term corresponds to a weighted least-squares approach. The first term corresponds to the pre-factor in the probability density function for a multivariate normal distribution; it is included because the variance itself is considered unknown. Multiple priors are also introduced to inform on model fitting. The device effects are realizations of an unbiased, normal random variable, such that

$$\ln p_{\rm pr} \left( l_k \mid s_1 \right) = -n_j \ln \left( s_1 \right) - \frac{1}{2} \sum_j \left( \frac{l_j}{s_1} \right)^2 + C_0 , \qquad (17)$$

where  $n_j = 5$  corresponds to the number of devices in this study. Priors on the three variances in  $\theta$  were each taken as Jeffreys priors,

$$\ln p_{\rm pr}(\sigma) = -2\ln\sigma \ . \tag{18}$$

Finally, a Gaussian prior was also applied to the  $\beta$ ,

$$\ln p_{\rm pr}(\beta) = -\frac{1}{2} \left(\frac{\beta}{\hat{\sigma}_{\beta}^2}\right)^2 + C_0 , \qquad (19)$$

where  $\hat{\sigma_{\beta}} = 3.0$  pg is approximately double the estimated value of  $\beta$ , only loosely constraining the value. As  $C_0$  is a constant throughout these expressions, explicit knowledge of  $C_0$  is not required to maximize the log-posterior. These log-priors were combined additively to form the overall log-prior. This procedure was applied to the data resulting from a 10 s averaging interval before being validated for other sampling intervals.

#### 3.3. Result of model fitting


Figure 4a shows a plot of the residual between the model, Eq. (9), and measurements, standardized by the variances inferred during model fitting. The residuals are reasonably consistent and normally distributed over the Δm<sub>c</sub> domain, indicating that the error model sufficiently describes the data. Systematic differences between the devices are not present after model fitting, indicating that these effects were estimated robustly. After standardization, measurements are roughly Gaussian, which is consistent with the treatment in the model (i.e., the use of a Gaussian likelihood) and makes statements about uncertainty simpler (e.g., as errors are symmetric and can be robustly summarized with a single standard error). Figure 4b to Figure 4f show that the trends seen in Figure 4a remain true even when the data are translated to longer sampling intervals of 20 s up to 1.5 minutes, using Eq. (8). Table 1 shows the corresponding values of the model fit to the measurements. The mean bias between the aethalometers and the reference was inferred to be β = 2.50 μg/m<sup>3</sup>.

Figure 4. Residual between the model and aethalometer measurements as a function of CERMS mass concentration, normalized by the estimate error for each point (i.e., standardized). Data correspond to the combined set of measurements for an averaging interval of 10 s, 20 s, 30 s, 1 min, and 1.5 min, with the model fit to the measurements using the 10 s averaging interval. Measurements are thinned by a factor of 5 for visualization. (b – f) Binned standardized residuals, stratified by averaging interval, demonstrating model adequacy across a range of averaging intervals.

Table 1. Values of the various effects and variances, alongside their uncertainties for the IR channel of the aethalometers. Device effects (e.g., I<sub>1</sub>) and the inter-device errors, I<sub>i</sub>, are multiplicative, such that they are stated as percentages of the mass concentration. Uncertainties in the variables are determined using a Markov chain Monte Carlo (MCMC) sampling procedure. These values are applied in Eq. (20) to estimate uncertainties in future aethalometer measurements.

| Variable                                | Bias                         | Device 1 bias | Device 2 bias             | Device 3 bias             | Device 4 bias             | Device 5 bias             | Inter-device variance     | Gaussian term | Poisson<br>term |
|-----------------------------------------|------------------------------|---------------|---------------------------|---------------------------|---------------------------|---------------------------|---------------------------|---------------|-----------------|
| Symbol                                  | $\beta \ [\mu \text{g/m}^3]$ | $l_1$ [%]     | <i>l</i> <sub>2</sub> [%] | <i>l</i> <sub>3</sub> [%] | <i>l</i> <sub>4</sub> [%] | <i>l</i> <sub>5</sub> [%] | <i>s</i> <sub>l</sub> [%] | y [pg]        | <i>p</i> [pg]   |
| Value                                   | 2.50                         | -15.2         | +0.9                      | +17.0                     | +10.8                     | -8.1                      | 10.0                      | 24.2          | 4.91            |
| Standard error of value $(k=1)$         | 0.03                         | 0.3           | 0.3                       | 0.3                       | 0.3                       | 0.5                       | 4.2                       | 0.3           | 0.12            |
| Expanded CoV of value $(k=2)^{\dagger}$ | 2.4%                         | -             | -                         | -                         | -                         | -                         | 84%                       | 2.4%          | 4.9%            |

†Expanded CoV are not stated for device effects, as values are distributed about zero, such that a CoV is not a reasonable representation of the uncertainties.

Figure 5 shows the difference between the CERMS and aethalometer mass concentration for a single device, with the goal of showing heteroskedasticity in the repeatability for a single device, without added inter-device contributions. Repeatability is derived from the assumed Poisson-Gaussian model, which, upon substitution of the values from Table 1, yields

$$s_{r} = \sqrt{6p \cdot \Delta t \cdot \Delta m + 6\gamma^{2} \cdot \Delta t} = \sqrt{29.5 \cdot \Delta t \cdot \Delta m + 3512 \cdot \Delta t} ,$$
(20)

where  $s_r$  and  $\Delta m$  are in pg,  $\Delta t$  is in min, and the prefactors of 6 result from the use of a 10 s (i.e., 1/6 min) reference sampling interval. This Poisson-Gaussian model reasonably describes the noise in the measurements, with an expanding interval as the

mass concentration increases due to Poisson contributions and a constant (Gaussian) contribution. The errors do not expand rapidly enough to be considered multiplicative, further validating model treatment.

Figure 5. Difference in mass measured by the aethalometer and predicted from the reference (CERMS) for a single device after correcting for the device-specific error. Green lines show the model fit (at zero, as this is a residual) and predicted error for a single device.

The inter-device variability is

$$s_{\rm L} = s_{\rm l} \cdot \Delta m = 0.1 \cdot \Delta m \tag{21}$$

or an expanded uncertainty of 20 % (k = 2) of the nominal value of mass change. Note, however, that the uncertainty in this value was substantial, due in part to the limited number of devices considered in the study. These errors were only visible for the highest mass concentrations. Figure 6 shows the residual in the measurements, without accounting for the individual device biases, for the highest CERMS mass concentrations ( $\Delta m_c > 600$  pg). Here, clustering is clear, though the repeatability was sufficiently large to cause overlap in the measurement ranges for each device. The magnitude of the inter-device variability is a bit larger than that observed by Cuesta-Mosquera et al. (2020) for AE33 aethalometers measuring laboratory samples, who saw differences in slope of around 12 % (k = 2) for the IR channel. The variability is similar to the spread observed by Chakraborty et al. (2023) for MA350 micro-aethalometers measuring ambient (traffic and wildfire) pollution. Both of those studies used AE33 aethalometers as reference devices, whereas this study used an absolute mass measurement.

290

285




Figure 6. Normalized device residual, that is the residual after removing offset and normalizing by CERMS mass concentration, resolved on a per-device basis for measurements made at CERMS mass concentrations above 600 pg. Inferred magnitude of the device effects are shown as horizontal rules.

The reproducibility is taken as the sum, in quadrature, of the repeatability and inter-device variance:

$$s_{\rm R} = \sqrt{0.01 \cdot \Delta m^2 + 29.5 \cdot \Delta t \cdot \Delta m + 3512 \cdot \Delta t} \tag{22}$$

where  $s_R$  and  $\Delta m$  are in pg and  $\Delta t$  is in min. Note that the Gaussian (or attenuation correction) contributions to the uncertainties increase as the sampling interval is increased but do not depend on the mass collected in the given sampling interval. The result is a Poisson-Gaussian-multiplicative error model (cf., Sipkens et al., 2017). The corresponding expanded (k = 2), reproducibility coefficient-of-variation (CoV) or relative standard deviation is

$$U_{\rm R} = 2\sqrt{0.01 + \frac{29.5 \cdot \Delta t}{\Delta m} + \frac{3512 \cdot \Delta t}{\Delta m^2}} \ . \tag{23}$$

where  $U_R$  is dimensionless,  $\Delta m$  is in pg, and  $\Delta t$  is in min.

#### 3.3.1. Uncertainty in mass concentration

The eventual quantity-of-interest for reporting is the mass concentration. Combining Eqs. (3) and (22), the corresponding uncertainty in the mass concentration is

$$s_{\rm R,M} = \sqrt{0.01 \cdot M^2 + \frac{29.5 \cdot M}{Q_a} + \frac{3512}{Q_a^2 \cdot \Delta t}} , \qquad (24)$$

where M and  $s_{R,M}$  are in units of  $\mu g/m^3$ ,  $Q_a$  is in mL/min, and  $\Delta t$  is in min. Restated as an expanded (k=2) CoV,

$$U_{\rm R,M} = 2\sqrt{0.01 + \frac{29.5}{M \cdot Q_{\rm a}} + \frac{3512}{\left(M \cdot Q_{\rm a}\right)^2 \cdot \Delta t}} \ . \tag{25}$$

where  $M \cdot Q_a$  always appear together. Again, this expression should still hold as the sampling interval and flow rate are changed. Though changes in flow rate were not explicitly validated in this study, our experimental design resulted in very different mass concentrations at different times and therefore did include variability in the product  $M \cdot Q_a$ .

## 3.4. Strategies to reduce uncertainty

The error model in this work describes two fundamentally different types of uncertainties, which require different approaches to mitigation: (*i*) inter-device biases and (*ii*) random errors or noise.

#### 3.4.1. Calibration of inter-device biases

The inter-device biases correspond to a fix bias for each device, which cannot be averaged away but can be effectively eliminated using regular correction of the device bias. In the limit where any drift in the device is insignificant between calibrations, the mass concentration can be given by

$$M = \left(1 - l_j\right) M_0 , \qquad (26)$$

where  $M_0$  is the mass concentration prior to accounting for the device specific bias. In this scenario, the uncertainties would be directly reduced to those as if  $s_1 = 0$ :

$$S_{R,M} = \sqrt{\frac{6p \cdot M}{Q_a} + \frac{6\gamma^2}{Q_a^2 \cdot \Delta t}}$$
 (27)

and


$$U_{\rm R,M} = 2\sqrt{\frac{6p}{M \cdot Q_{\rm a}} + \frac{6\gamma^2}{\left(M \cdot Q_{\rm a}\right)^2 \cdot \Delta t}} \quad . \tag{28}$$

In the absence of knowledge of the precise device bias, one should revert to the previous expressions.

# 320 3.4.2. Averaging for noise reduction

The remaining errors are random and can thus be reduced using averaging techniques, under the assumption that the underlying mass concentration is relatively constant in each averaging period. Plotting Eq. (25) as a function of sampling time


in Figure 7 yields a plot, akin to an Allan-Werle plot, albeit without contributions from device drift (drift could not be assessed in this study as the mass concentration was not held constant, making it difficult to distinguish changes in the mass concentration from drift in the instrument). At the bottom of this plot is the inter-device limit, corresponding to the uncertainties between devices that cannot be removed by way of averaging, thus representing a natural lower limit. Increasing the sampling interval reduces the uncertainties as expected, but with limitations due to amplification of Poisson noise as the sampling interval is increased.

Figure 7. The reduction in the expanded (k = 2) coefficient of variation (or relative standard deviation) of the mass concentration as the sampling interval increases. Curves correspond to different values of  $M \cdot Q_a$ , labelled with in [pg/min] in bold blue, with the corresponding mass concentration [µg/m³] at a flow rate of 75 mL/min, representative of this study, indicated in green italics.

In the upper region of Figure 7, uncertainties expand beyond the value of the mass concentration itself. Generously defining the limit-of-detection (LoD) as the point at which two times (so, k = 2) the reproducibility standard error is equal to the measurement (McNaught et al., 1997), that is when the signal-to-noise ratio is 2, an expression for the LoD can be derived by solving the quadratic equation that results by setting a value for  $s_{R,M}$  in Eq. (24). Specifically,

$$LoD \approx \frac{-6p - \sqrt{36p^2 - 24(s_1^2 - 1/4)\gamma^2/\Delta t}}{2Q_a(s_1^2 - 1/4)},$$
(29)

where  $Q_a$  is in mL/min and  $\Delta t$  is in min. Note that, as the attenuation coefficient contributes significantly to the uncertainties in the measurements, one would expect that Figure 7 and the associated expressions to be specific to the dual spot correction applied in this work. In fact, this noise source is likely an excellent target for reducing the overall uncertainties in the


measurements. Despite this fact, the functional form used for the Poisson and inter-device errors are expected to be more general and to apply to similar devices.

Since the attenuation measurement of the aethalometer can be monitored at high frequency, the ideal sampling interval could be updated dynamically, to always maintain a reasonable uncertainty. This approach has already been demonstrated by the aethalometer smoothing algorithm of Hagler et al. (2011). Our work extends the Hagler algorithm in two ways. First, in addition to smoothing, our algorithm estimates corresponding uncertainties. Second, our algorithm includes an input parameter in the form of a desired uncertainty, based on which the smoothing amount is deduced. Under the assumption of constant frequency measurement and constant aerosol flow rate, the accumulated mass can be equally stated as

$$\Delta m = \Delta t_0 \cdot Q_a \cdot \sum_{i=1}^n M_i = n \Delta t_0 \cdot Q_a \cdot \overline{M} , \qquad (30)$$

where  $M_i$  denotes the *i*th measurement of mass concentration,  $\Delta t_0$  is the original sampling interval for M prior to averaging, and n is the number of intervals used for averaging. Now, the uncertainties in the average M are

$$S_{R,\overline{M}} = \sqrt{S_1^2 \cdot \overline{M}^2 + \frac{6p \cdot \overline{M}}{n \cdot Q_a} + \frac{6\gamma^2}{\left(n \cdot Q_a\right)^2 \Delta t_0}}$$
(31)

and


$$U_{\rm R,\bar{M}} = 2\sqrt{s_{\rm l}^2 + \frac{6p}{n \cdot \bar{M} \cdot Q_{\rm a}} + \frac{6\gamma^2}{\left(n \cdot \bar{M} \cdot Q_{\rm a}\right)^2 \Delta t_0}} \ . \tag{32}$$

With knowledge of the bias for a specific device, these expressions would be reduced by setting  $s_1 = 0$ , as before. Code to perform this procedure – averaging until a specific error is reached – is provided in the online supporting information. The output is compared to the output of the algorithm by Hagler et al. (2011). The two approaches can be made consistent depending on the value of the change in attenuation parameter (for the Hagler algorithm) and the desired uncertainty (for the current averaging algorithm), as shown in Figure 8. The algorithm only requires the attenuation data to automatically determine when the filter has changed.

Figure 8. The effect of different averaging of filtering approaches to reduce random errors in the measured eBC, applied for one of micro-aethalometers used in this work. Grey points indicate the eBC reported by the device, while solid lines include single processed by (a) the algorithm proposed by Hagler et al. (2011), (b) averaging to a specified repeatability is reached (here,  $U_{R,M} = 10$  %), and (c) applying a Kalman filter informed by the error model in this work.

The uncertainty expressions in this work can also enable the use of Kalman filter approaches for post-processing. These approaches step through the signal, using an estimate at a previous point in time to inform on measurements at the current time while propagating uncertainties forward through time. Uncertainties are a direct output of the algorithm and vary depending on the mass concentration level. Overall, Figure 8 shows that the filtered mass concentrations are similar to those from the averaging approaches, albeit with finer temporal resolution. Errors tend to be smaller than the averaging approaches at higher mass concentrations and vice versa.

#### 4. Conclusions

Experiments were performed to compare five different aethalometers with reference mass concentrations generated using a CPMA-electrometer reference mass standard (CERMS). Device effects were multiplicative, consistent with other studies in the literature, while noise from the attenuation correction manifested as roughly Gaussian noise. While the dual spot correction algorithm was found to be effective in correcting biases in the measurements, the correction caused an increase in random errors in the measurements increased. A reduction in the multiplicative inter-device errors would reduce the overall uncertainties and reduce the minimum uncertainties achievable with the system, though the precise reason for the inter-device




360

385

differences was not determined in this study. The overall standard error in the reproducibility for the micro-aethalometers in this work is

$$s_{\rm R,M} = \sqrt{0.01 \cdot M^2 + \frac{29.5 \cdot M}{Q_a} + \frac{3512}{Q_a^2 \cdot \Delta t}} , \qquad (33)$$

where M is in  $\mu g/m^3$ ,  $Q_a$  in mL/min, and  $\Delta t$  in min. This expression includes inter-device uncertainties and can be used as an uncertainty for measurements of black carbon when using the micro-aethalometers considered in this work.

Two approaches are considered to reduce errors. First, if appropriate action can be taken to calibrate for device-specific biases, these uncertainties can be reduced to contributions from only the latter two terms in Eq. (33). Second, under the assumption of a slowly changing mass concentration, averaging can reduce the two remaining error terms. If both of these approaches are undertaken, the uncertainties would be reduced to

$$s_{R,\bar{M}} = \sqrt{\frac{29.5 \cdot \bar{M}}{n \cdot Q_a} + \frac{3512}{(n \cdot Q_a)^2 \Delta t_0}} . \tag{34}$$

Implementation of noise reduction algorithms using averaging and a Kalman filter for are included in the online supplemental information.

# Code availability

Code for averaging aethalometer data to reach a specified uncertainty and compare the result to the algorithm by Hagler et al. (2011) has been made available in the supplemental information. The code will also be made available on GitHub at <a href="https://github.com/tsipkens/aeth-app">https://github.com/tsipkens/aeth-app</a>.

## Data availability

Partially processed data is included alongside the manuscript.

## 395 Author contribution

TAS: methodology, formal analysis, and led the writing (original draft). JCC: data processing and contributed to writing (original draft and editing), and conceptualization. KC: performed laboratory measurements. LR: investigation, deployment and commissioning of the micro-aethalometers and preliminary data analysis. JA: funding acquisition, supervision, and comments on the manuscript. JSO: supervision, funding acquisition, and experimental design.

# 400 Competing interests

The authors declare that they have no conflict of interest.

#### Financial support

The authors would like to acknowledge funding from the Government of Canada's Environmental Damages Fund though the Climate Action and Awareness Fund (CAAF) and the Small Teams Initiative of the National Research Council of Canada.

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
