# Peer review of "Quantitative uncertainty and post-processing for microaethalometers measuring black carbon"

_EGUsphere, 2025_

## Referee Comment (RC1)

**Egusphere-2025-4209 - Review**

The article presents new statistical methods for determination of uncertainties connected to interinstrument differences and measurement noise and applies them to micro-aethalometers MA300 and MA200. The experiment was performed by comparing tested instruments with Centrifugal Particle Mass Analyzer-Electrometer Reference Mass Standard (CERMS). The article is well written with appropriate introduction and methodology and description of the results. There are two issues which need to be addressed.

Filter photometers suffer from particle size dependent response, which makes it important to use realistic particle size for testing. The soot particles selected by CERMS with a mobility diameter of 235 nm are larger compared to the diesel engine and wood stove emissions (Laborde et al., 2012) and are more comparable to wildfire emissions. Measurement of larger particles is expected to result in higher "quantum" noise (from discrete arrival of particles on the filter), as already observed by the authors. Particle size is expected to influence the instrument response to BC mass.

The second issue is the influence of the filter loading effect. Figure 2 show jumps in instrument response during tape advance which indicate the presence of filter loading effect. Same can be concluded by analyzing relative instrument response (eBC/ref\_mass) as a function of attenuation using data in the Supplement:

Authors should discuss the implications of filter loading effect, for example the increased variation of the instrument response.

**Line by line comments**

Page 1, Line 17: "A quantitative expression is provided for the uncertainty in the aethalometer measurements as a function of mass concentration, sampling interval, and flow rate."

It should be noted that these uncertainties present one part of the uncertainties connected to the filter photometers. Uncertainties connected to filter loading effect, particles size, cross-sensitivity to scattering and mass absorption cross-section can be higher than inter-instrument variability and noise.

Page 2., Line 42:

There have been more measurements of the influence of particle size on instrument response. It seems to be a general feature of filter photometers: it was observed for AE33, CLAP and MAAP (Ramshoo et al, 2022; Drinovec et al., 2022; Yus-Dies et al., 2025).

Ramshoo et al, 2022; https://doi.org/10.5194/amt-15-6965-2022-supplement

Drinovec et al., 2022; https://doi.org/10.5194/amt-15-6965-2022

Yus-Diez et al., 2025; https://doi.org/10.5194/amt-18-3073-2025

Page 4, Line 114: "size distribution had a geometric mean mobility diameter of 235 nm"

The selected particle size depends strongly on charge distribution from the unipolar charger. How stable was the charge distribution during the experiment?

Page 7. Fig 3.

In Figure 3. eBC measured by aethalometers is compared to particle mass concentration of CERMS. What is the contribution of organics to the sample mass? What would be the effect of thermodenuder or catalytic stripper?

Page 7. Line 158. "with structured artifacts as a function of attenuation appearing in the measurements when not correcting the data (cf. Figure 2b)"

This is in contrast with the caption of Figure 2 which indicates that the data is corrected for the filter loading effect (except for data where ATN<3). Please clarify which data is corrected for the filter loading effect.

Page 9, Line 202: "While not shown in Figure 3, it is noted that the attenuation coefficient has minimal effect above an attenuation of 3,".

Data on Figure 3 and in the supplement suggest that the filter loading effect is still present.

Page 18, Line 376: "While the dual spot correction algorithm was found to be effective in correcting biases in the measurements"

Please see the comments on filter loading effect above.

Page 19, Line 394. "Partially processed data is included alongside the manuscript."

Please provide both raw and processed data.